# Systematic Study of the Impact of Pulsed Electric Field Parameters (Pulse/Pause Duration and Frequency) on ED Performances during Acid Whey Treatment

**DOI:** 10.3390/membranes10010014

**Published:** 2020-01-11

**Authors:** Guillaume Dufton, Sergey Mikhaylin, Sami Gaaloul, Laurent Bazinet

**Affiliations:** 1Institute of Nutrition and Functional Foods (INAF), Dairy Science and Technology Research Center (STELA) and Department of Food Sciences, Université Laval, Québec, QC G1V 0A6, Canada; guillaume.dufton.1@ulaval.ca (G.D.); sergey.mikhaylin@fsaa.ulaval.ca (S.M.); 2Laboratoire de Transformation Alimentaire et Procédés ÉlectroMembranaires (LTAPEM, Laboratory of Food Processing and ElectroMembrane Processes), Université Laval, Québec, QC G1V 0A6, Canada; 3Parmalat, Victoriaville, Québec, QC G6P 9V7, Canada; sami_gaaloul@parmalat.ca

**Keywords:** electrodialysis, pulsed electric fields, scaling, acid whey, demineralisation, lactic acid removal

## Abstract

Processing acid whey is still a challenge for the dairy industry due to its high lactic acid and mineral contents. Their removal processes represent a high investment and running cost in addition to significant production of polluting effluents. A previous study showed that the use of electrodialysis with the application of pulsed electric fields (PEFs) was sufficiently efficient to produce dryable acid whey with reduced scaling issues during the process. In the present work, eight PEF conditions using different pulse/pause durations and frequencies were tested for 1) process optimization and 2) understanding of the underlying mechanisms involved in PEF process improvements. Best results were obtained for PEF conditions (5 s/5 s) and (15 s/15 s) with almost complete scaling mitigation and minimal energy consumption (5.3 ± 0.4 Wh/g of lactic acid vs. 9.33 ± 1.38 Wh/g for continuous current). Longer pause times also led to better divalent ion demineralization at the expense of sodium elimination induced by stronger affinity with the membrane and longer retention times. For the first time, PEF parameters of relatively low frequencies (<1) were studied in sub-limiting current conditions on a complex solution such as acid whey.

## 1. Introduction

The dairy industry faces growing challenges regarding its by-product production and valorization. Over the last decade, the demand for fresh cheese and Greek style yogurt among other products has been greatly increasing. This has led to a large production of whey as it represents, during yogurt, cheese or casein transformation, up to nine tenths of the initial milk volume. Sweet whey, obtained from rennet cheese manufacture, is well known and valorized for its proteins and lactose contents [1,2,3] but acid whey valorization faces a major processing problem during drying [4]. The highly hygroscopic character of lactic acid and minerals such as calcium and magnesium drastically hinders the spray-drying process of acid whey, resulting in low process performance and clogging of the equipment [5,6]. In consequence, before drying, a separation step is necessary to get rid of the lactic acid and most of the minerals. This stage is usually performed through serial processes for an optimal outcome (e.g. ion-exchange resins, electrodialysis, and nanofiltration) but several studies have been recently conducted in order to reduce the economic and ecological cost of such processes [7,8,9,10]. By optimizing electrodialysis, the ion-exchange resin steps could be removed in part or fully, increasing the sustainability of the whole process.

Electrodialysis (ED) alone is capable of reaching sufficient separation rates for spray-drying during acid whey treatment but it is subjected, through conventional continuous current (CC) use, to major membrane scaling [11,12]. This membrane scaling, mainly composed of calcium phosphate [12,13], is caused by the establishment of alkaline pH conditions in the boundary layer of the anion-exchange membranes (AEMs) following water molecules’ dissociation. This water splitting phenomenon is itself brought about by the intensification of concentration polarization due to ion flux differences in the membranes, membrane boundary layers, and solutions [14]. Since fouling and/or scaling of the ion exchange membranes is one of the most important limiting factors in the use of ED, several solutions have been explored. In the recent years, scaling and whey protein fouling were successfully controlled and mitigated by the application of pulsed electric fields (PEF) in a number of studies treating model salt solutions [15,16,17] and more recently scaling during sweet and acid whey demineralization [13,18]. The application of PEF consists of the use of a current regime with alternating pulse and pause periods of set amounts of time. The presence of pause lapses allows the concentration polarization phenomenon to be drastically reduced, thus mitigating scaling reactions in addition to enhancing ion separation [19,20]. However, the impact of modifying the pulse and/or pause lapse of a certain duration (from 0 to 20 s) during ED-PEF has never been systematically studied and reported.

In this context, this study aimed to optimize the treatment of acid whey by ED with the use of PEF, as well as gain an understanding of membrane scaling and ionic migration behavior depending on the following PEF parameters: pause duration (15, 25, and 35 s), pulse duration (15, 25, and 35 s), and frequency (0.1, 0.03, 0.02, and 0.14 Hz). The study statistical design was a surface response experiment, centered on the condition 25 s/25 s according to Dufton et al. [13]. Eight conditions (5 s/5 s, 15 s/15 s, 15 s/25 s, 25 s/25 s, 25 s/15 s, 25 s/35 s, 35 s/35 s and 35 s/25 s) were studied.

## 2. Materials and Methods

### 2.1. Acid Whey

The raw product was provided by the Parmalat-Canada (Winchester, Ontario, Canada) processing plant. The acid whey was transported at 4 °C and stored in 1 L aliquots at −30 °C. Each ED run used 2 L of acid whey that were thawed beforehand at 4 °C. The whey composition is described and compared to other studies in Table 1.

### 2.2. Electrodialytic Configuration

The selected configuration (Figure 1) consisted of two separation units using the stacking commonly used in the dairy industry or lactic acid removal processes and in recent studies on acid whey treatment [7,13,22]. Electrodialysis experiments were performed using an MP type cell (ElectroCell AB, Täby, Sweden) with an effective surface area of 100 cm^2^. The anode was a dimensionally stable electrode (DSA-O_2_) while the cathode was a stainless steel electrode. The potential difference was generated by a power supply (Model HPD 30–10, Xantrex, Burnaby, BC, Canada) and the PEF was applied thanks to a Pulsewave^TM^ 760 Switcher (Bio-Rad Laboratories, Richmond, CA, USA). The solutions circulated using centrifugal pumps (Baldor Electric Co., Fort Smith, AR, USA) and the flow rates were controlled by flow meters (Aalborg Instruments and Controls, Inc., Orangeburg, SC, USA).

The ED configuration consisted of the stacking of five commercial food grade cation and anion exchange membranes (respectively CEM and AEM). Diluate (acid whey compartment: AWComp) and concentrate (organic acid recovery compartment: OAComp) solutions were circulated between the membranes defining three closed loops. The solutions used were a 20 g/L Na_2_SO_4_ electrolyte solution (volume of 2 L, flow rate of 4 L/min), a 5.5 g/L NaCl aqueous concentrate solution (2 L, 4 L/min), and acid whey (2 L, 4 L/min). To ensure a continuous recirculation, external tanks containing the solutions were connected to each closed loop. The closest CEM to the cathode (C3, see Figure 1) was added in order to avoid any anion migration in the electrolyte solution compartment, specifically lactate.

### 2.3. Protocol

All ED treatments were carried out with a similar driving force to that used in previous studies and a constant current density of 100 A/m^2^ was applied. This current density represented, at the beginning of the experiment, 44% of the limiting current density determined for acid whey according to Cowan and Brown method [23]. Taking a previous preliminary study as the basis [13], the PEF condition with a pulse/pause combination of 25 s/25 s was chosen as the control condition as it demonstrated the best results in term of scaling mitigation. From it, a surface response experiment with seven other conditions was designed with the objective of assessing the impact of different PEF parameters during acid whey electrodialysis. The considered PEF parameters were pause duration, pulse duration, and frequency. The treatments tested are reported in Figure 2. The experiment durations were calculated according to the pulse/pause combination applied for all conditions to correspond to an effective treatment of three hours or to a number of charge transported of 10,800 C. The solution tanks were kept at room temperature around 20 °C and three replicates were performed for each current condition. At the end of each run, before dismantling the cell, the whole system (tanks, tubing, and ED cell) was rinsed for 5 min with water to remove all superficial or non-adsorbed scaling.

Before and after each ED, the raw acid whey minerals, lactic acid, and lactose contents were analyzed. Membrane thickness and electrical conductivity were also measured before and after each run and the membranes were dried and kept for mineral analysis. Electrical conductivity and pH for each solution, as well as applied voltage were recorded every 10 min of effective current application (or 600 C). Organic acid concentration was determined by high-performance liquid chromatography (HPLC) on samples taken at 0, 30, 60, 120, and 180 min of effective current application (corresponding to 0, 1800, 3600, 7200, and 10,800 C) in both AWComp and OAComp.

### 2.4. Analyses

All analyses were performed on at least three technical samples.

### 2.5. Total Solids and Ash Contents

According to the Association of Official Analytical Chemists (AOAC) methods 990.20 and 945.46, raw acid whey samples were weighed before drying for one hour on a heating plate (Corning PC-420 Hot Plate Stirrer, NY, USA). The dried samples were then weighted again for total solid content determination and further ashed in a furnace at 550 °C overnight until they turned white. The samples were weighed after cooling and the ash content was determined as follows (m refers to the measured weights):(1)$$ash\;\%=\frac{100\left(m_{crucible+ashes}-m_{crucible}\right)}{m_{crucible+sample}-m_{crucible}}$$

### 2.6. Mineral Concentration

Calcium, potassium, magnesium, sodium, and phosphorus concentrations were determined by optical emission spectrometry with an inductively coupled plasma as atomisation and excitation source (ICP-OES Agilent 5110 SVDV Agilent Technologies, Victoria, Australia), using the following wavelengths: 393.366; 396.847; 422.673 (Ca), 766.491 (K), 279.553; 280.270; 285.213 (Mg), 588.995; 589.592 (Na), 177.434; 178.222; 213.618; 214.914 (P). The analyses for all ions were carried out in axial and/or radial view, directly on acid whey samples diluted 20 times, then 10 mL samples diluted in distilled water were used for element concentration determination in g/L.

### 2.7. Organic Acid Concentration

Organic acid concentrations were determined by high-performance liquid chromatography (HPLC) using a chromatograph from Waters (Waters Corp., Milford, MA, USA), equipped with a Hitachi (Foster City, CA, USA) differential refractometer detector L-7490. An ICSep ICE-ION-300 column (Transgenomic, Omaha, NE, USA) was used with 8.5 mM of H_2_SO_4_ (180 µL H_2_SO_4_/L) as the mobile phase and at a flow rate of 0.4 mL/min. The column temperature was kept constant at 40 °C. Samples were centrifuged for five minutes at 5000 rpm (Allegra™ 25R Centrifuge, Beckman Coulter, Brea, CA, USA) and filtered (0.22 µm nylon; CHROMSPEC Syringe Filter, ON, Canada) before injection (15 µL). A mixture of lactose anhydrous (PHR1025), citric acid (251275), DL-lactic acid (L1750), and acetic acid (338826) (from Sigma-Aldrich, St. Louis, MO, USA) was used as an external standard to perform the quantification in g/L.

### 2.8. pH

The pH of acid whey (AWComp) and organic acid recovery (OAComp) solutions were measured using a pH meter model SP20 (VWR Symphony, Thermo Orion West Chester, PA, USA).

### 2.9. Conductivity

An YSI conductivity meter (Model 3100, Yellow Springs Instrument, Yellow Springs, OH, USA) equipped with an immersion probe (Model 3252, cell constant K = 1 cm^−1^) was used for measuring values in acid whey (AWComp) and organic acid recovery (OAComp) solutions. Mineralization (MR) and demineralization (DR) rates are presented in Equations (2) and (3).
(2)$$MR=\left(1-\frac{OAComp\;conductivity\;at\;time\;t=0}{OAComp\;conductivity\;at\;time\;t}\right)\times 100$$
(3)$$DR=\left(1-\frac{AWComp\;conductivity\;at\;time\;t}{AWComp\;conductivity\;at\;time\;t=0}\right)\times 100$$

### 2.10. Global System Resistance

The global system resistance (R, in Ω) was calculated according to Ohm’s law (R = U/I). The voltage (U, in V) and current intensity (I, in A) values were directly obtained from the power supply.

### 2.11. Relative Energy Consumption (REC)

(4)$$REC=\frac{\int_{t=0}^{t=end}\frac{U\times I}{3600}dt}{m_{lact.acid}}$$
where REC is the relative energy consumption (in Wh/g of lactic acid transported), U—the voltage applied (in V), I—the applied current density (in A), t—the effective time duration (in s), and m_lact.acid_ is the total mass of lactic acid at the end of treatment in the OAComp (in g). The effective time duration for PEF conditions only took into account the pulse periods.

### 2.12. Membrane Mineral Concentration

The same elemental concentrations as for the liquid solutions were measured for the membranes by ICP-OES as described previously. For each configuration, the analysis was conducted in triplicate on 50.41 cm^2^ pieces of AEM (C**A**CAC the bold character highlights the membrane analyzed) and CEM (CA**C**AC). Pristine AEM and CEM were also analysed as controls. Membrane pieces of 50.41 cm^2^ were cut, weighed, and dried at 60 °C overnight in an oven (VWR Gravity Convection Oven, Radnor, PA, USA). The dried samples were then ashed in a furnace at 550 °C overnight until they turned white. The samples were weighed after cooling and the ash content was determined according to Equation (1). The ashes were resolubilized in 1 mL 25% nitric acid and diluted in 50 mL total volume with demineralized water (PURELAB® Ultra, ELGA, High Wycombe, UK) before ICP-OES analysis.

### 2.13. Statistical Analyses

Analyses of variance (ANOVA) were performed on data and Tukey as well as Dunnett tests (α = 0.05 as probability level) were used to compare the eight treatments between themselves and with the PEF control condition (25 s/25 s), respectively (SigmaPlot software, version 12.0 for Windows, MilliporeSigma, Burlington, MA, USA). 

## 3. Results

### 3.1. Whey and Recovery Solution Analysis

#### 3.1.1. Lactate Migration

The statistical analysis did not show any difference between the eight treatments regarding lactate migration in both AWComp (P = 0.07) and OAComp (P = 0.67). As shown in Figure 3a, lactate concentration decreased from an average 7.68 ± 0.26 g/L to an average 4.26 ± 0.16 g/L in the AWComp and increased accordingly to reach 3.48 ± 0.17 g/L in the OAComp (Figure 3b). These results correspond to a lactic acid removal and recovery rates of 44.5 ± 1.2% regardless of the PEF condition tested. However, among the eight conditions, even if there was no significant difference, it can be mentioned that the 35 s/35 s PEF condition reached the lowest lactic acid recuperation of 3.22 ± 0.19 g/L while the 15 s/15 s and 25 s/15 s obtained the highest one with an average 3.59 ± 0.41 g/L.

During current application, the whey lactic acid conjugate base, lactate, migrated through the AEMs to reach the OAComp. For an equivalent number of charges transported, each PEF condition reached a similar lactic acid removal rate. These results are similar to the ones obtained for the 25s/25s PEF condition in a previous study [13] representing a 16% gain in comparison to the application of CC in the same conditions. However, no difference can be seen between the PEF conditions tested in this experiment. The advantages of using PEF treatments on ion migration enhancement are related to the reduction of the concentration polarization during the pause periods, thus improving the ion availability at the membrane’s boundary layers when the current is switched on again [20,24]. Therefore, the mechanisms involved occur during the extremely short period between the pause duration and the current restoration. Our conditions using relatively long periods of pause and pulse do not take full advantage of these phenomena and might be too similar for a difference in lactic acid migration to be visible. 

#### 3.1.2. pH

In the AWComp the pH decreased during ED from an average 4.43 ± 0.01 to 4.04 ± 0.03 (Figure 4) after 3 h of effective current application with no statistical difference between the eight different PEF conditions (P = 0.13). Among the conditions tested, the 15 s/25 s one had the lowest delta pH while the conditions 25 s/15 s and 35 s/25 s had the highest. Regarding the OAComp, the pH followed the same pattern for all conditions: it decreased until reaching a minimum value before a stationary state followed by an increasing phase around 5400 C (see Figure 5a). For all conditions, the initial pH was at an average 6.84 ± 0.08 and reached a final average of 6.07 ± 0.08 after 3 h of effective current application without statistical difference between conditions (P = 0.26). The condition with the lowest late pH increase was the 15 s/25 s condition. By only considering conditions assessing the pulse duration variation effect during PEF-ED, even if there was no statistical difference between their delta pH (P = 0.08), a tendency was observable: the longer the pulse time, the higher the final pH in the OAComp (Figure 5b).

In the AWComp, the pH decreased due to organic acids dissociation and potentially, at some point during treatment, to water splitting occurring on the diluate side of the AEM surface [12,25]. Indeed, the pH variation may be explained by dissociation of, amongst others, lactic acid following lactate migration, since protons were released into the depleted solution when they entered the AEM and by water dissociation catalyzed by AEM ionogenic groups and by weak-acid anions [18]. The pH variation was similar for all PEF conditions tested with a delta pH between 0.35 and 0.42 which is lower than the variation reported for the use of CC in the same experimental conditions [13]. As the lactic acid separation was slightly enhanced, this reduction in pH variation might be an indicator of the water splitting phenomenon reduction thanks to the use of PEF. Regarding the OAComp, the pH decreased due to protons migrating from the acid whey and then stabilized before slowly increasing again because of water splitting. This was described in previous studies [10,13] and the effect of PEF application on the reduction of pH variations is here confirmed by the slow and low pH increase during the last phase of ED treatment [15]. The eight PEF parameters tested here did not present significant differences in terms of pH variations. The chosen conditions might be too similar to induce clear modifications in solution properties. Nevertheless, tendencies can be observed such as the later inflection point between 5400 C and 6600 C depending on the PEF condition or the rising pH final value with the pulsation duration increase (Figure 5b). This observation can be related with the occurrence of the water splitting phenomenon during the application of electrical current. The longer this application, the higher the appearance and intensity of the water dissociation phenomenon.

#### 3.1.3. Conductivity and Mineral Concentrations

Regarding demineralisation, solution conductivity of the AWComp shown in Figure 6a decreased from an average of 7.81 ± 0.14 mS·cm^−1^ to 2.50 ± 0.05 mS/cm for all conditions without statistical difference (P = 0.68) after 3 hours of current application. Accordingly, conductivity in the OAComp rose from an average of 8.17 ± 0.15 mS/cm to 14.06 ± 0.23 mS/cm with no significant difference between the eight PEF conditions (P = 0.92) (Figure 6b). The final conductivity values corresponded to demineralisation and mineralisation rates of 68.0 ± 0.5% and 75.5 ± 0.9%, respectively. In addition, ICP-OES analysis allowed determination of the specific concentrations of minerals in the AWComp before and after ED treatment as reported in Figure 7. Calcium averaged initial concentration in acid whey was 0.93 ± 0.05 g/L and it decreased for all conditions after ED treatment. As shown in Figure 7a, the eight PEF conditions reached a similar separation rate of 61.1 ± 1.1% except for the 25 s/15 s and the 25 s/35 s conditions (P = 0.01) which achieved 55.2 ± 3.7% and 65.4 ± 1.7% of calcium separation, respectively. A representation of conditions related to the effect of pause duration variation in Figure 7b allowed the observation of a significant enhancement of calcium migration with an increasing pause duration (P = 0.01). From the 25 s/15 s to the 25 s/25 s condition, calcium migration was improved by 6.5% and from the 25 s/25 s to the 25 s/35 s condition, it was improved by 11.1%. Concerning pulsation duration and frequency variations in the PEF conditions tested, there were no significant differences between conditions on all mineral migrations. Regarding magnesium concentration in Figure 7c, the initial average value was 0.10 ± 0.00 g/L and just like calcium, all PEF conditions reached a similar demineralization rate of 45.0 ± 1.7% except for the same two conditions, 25s/15s and 25s/35s, which reached 37.7 ± 5.3% and 51.2 ± 1.3%, respectively. By only taking PEF conditions involving pause duration variations (Figure 7d), a migration enhancement identical to that of calcium was observed with longer pause durations (P < 0.001). From the 25 s/15 s to the 25 s/25 s condition, magnesium migration was improved by 9.5% while increasing by 18% between the 25s/25s and the 25 s/35 s condition. In the case of potassium (Figure 7e), the averaged initial concentration was 1.42 ± 0.04 g/L. This decreased after ED treatment with an averaged demineralization rate of 83.1 ± 1.5% for all conditions (P = 0.25). The lowest demineralization rate was achieved by the PEF condition 25s/35s which was significantly different from the other two conditions (P = 0.03) when only considering pause duration variations related ones (Figure 7f). Sodium concentrations, as shown is Figure 7g, had an initial value, before ED, of 0.43 ± 0.02 g/L. After treatment, three conditions were statistically different from the majority (P < 0.001). While the averaged demineralization rate was 52.6 ± 3.9%, the PEF conditions 15 s/15 s, 15 s/25 s and 25 s/35 s reached lower demineralization rates of 46.2 ± 3.1%, 34.7 ± 1.0% and 31.2 ± 1.3%, respectively. The observation of conditions related to pause duration variations in Figure 7h allowed to highlight a significant effect of a higher pause duration on the reduction of sodium migration. Sodium demineralization decreased by 9.4% and 40.7% from 25 s/15 s to 25 s/25 s conditions and from 25 s/25 s to 25 s/35 s conditions, respectively. Regarding phosphorus (data not shown), the initial concentration of 0.68 ± 0.03 mg/L was reduced by 59.1 ± 1.2% after treatment for all conditions (P = 0.88). 

Total demineralisation and mineralisation rates represented here by each solution conductivity measurement are not exactly similar. This might be due to the contribution of ions produced by water splitting at some point during treatment. Thus, conductivity measurement might not be the appropriate method to determine precise demineralisation rates during our experiments. However, similar results have been found using specific mineral ions measurements by ICP-OES with a global demineralisation (including calcium, magnesium, sodium, potassium, and phosphorus) of 67.3 ± 1.1%. Regarding specific migration of mineral ions, calcium and magnesium have similar behavior depending on the PEF conditions applied. By increasing the pause duration, their separation from the acid whey was proportionally enhanced, however, this seemed to exponentially hinder sodium separation. The influence on potassium migration was not as clear but longer pause durations might lead to the same mitigation. This difference in monovalent and divalent ion demineralization proportions caused by an increased pause duration could be explained by several points reported in the literature. First of all, it has been shown that low current density will favor divalent ion migration due to their stronger interaction with the membrane’s functional groups [26,27]. Inversely, higher current density will increase the concentration polarization and favor the separation of monovalent ions thanks to their higher diffusivity in the membrane’s boundary layer. During pause duration, the diffusion layer is rapidly depleted and the concentration polarization reduced but a potential difference is preserved and a slight electro-osmotic movement towards and through the membranes still persists in solution [19]. Hence, divalent ions such as calcium and magnesium will preferentially interact with the membrane’s ion-exchange groups due to their higher sorption coefficient [28]. Calcium and magnesium also possess higher retention times leading to them monopolizing most of the negatively charged ion-exchange groups even after electrical current restoration [29]. Consequently, the longer the pause duration, the stronger the competition between calcium/magnesium and sodium/potassium at the membrane’s ion-exchange groups. Concerning the fact that no significant differences between conditions were observed for pulse duration and frequency variations in the PEF conditions, this was the first time it has been reported. In the literature, by increasing pulse duration or decreasing PEF frequency, the advantage of the diffusion layer depletion during the pause lapses vanishes rapidly leading to lower transfer rates [20]. However, this study was performed using lower frequency PEF conditions, a complex solution, and importantly, solution flow that could have altered ion transfer.

### 3.2. Global System Resistance and Relative Energy Consumption

The initial global system resistance averaged value was 10.4 ± 0.2 Ohm. After 3 h of effective current application, there was no statistical difference between the eight PEF conditions tested (P = 0.23) due to high standard deviation values mainly caused by the presence of uneven membrane scaling. The increase in resistance was mainly due to the demineralization of the solution coupled to the presence of scaling. However, all conditions still had a lower final resistance than the one reached by applying CC (42.0 ± 6.69 Ohm) in a previous study [13] (Figure 8a). This reduction in comparison to the continuous current was the lowest for the 35s/35s condition with a 19.4% final decrease while the 15s/15s condition allowed a reduction of 40.1%. The highest final system resistance was reached for conditions 35 s/35 s and 35 s/25 s with 33.8 ± 5.1 Ohm and 33.5 ± 5.3 Ohm, respectively while the lowest one was achieved by the 15 s/15 s condition with 25.2 ± 2.7 Ohm. By separately observing the PEF conditions involving pause duration and pulse duration variations in Figure 8b,c, respectively, it seems that longer pause durations had a favorable impact on the system resistance reduction while longer pulse times had the opposite effect. 

Regarding the relative energy consumption, Table 2 regroups the average consumption of all PEF conditions in decreasing order. Just like global system resistance, no statistical difference was observed between the eight conditions (P = 0.36). However, the increasing pulse duration represented by the 15 s/25 s, 25 s/25 s, and 35 s/25 s conditions appeared to induce higher energy consumption for equivalent current application duration with 5.6 ± 0.3 < 5.8 ± 0.3 < 6.7 ± 1.0 Wh/g, respectively.

The inflection point in the global system resistance evolution around 6000 C depended on the PEF condition and was correlated with the beginning of pH variation observed in the OAComp that was attributed to the apparition of water splitting phenomenon. Such an effect of water dissociation on system resistance is related to the apparition of membrane scaling coupled with the rather high demineralisation rate reached at this point [13,30,31]. The use of PEF is known to mitigate scaling formation during model solutions ED [15,32] and here, according to the differences observed between the eight PEF conditions, shorter pulse durations with either similar or longer pause durations seemed to decrease both final electrical resistance and relative energy consumption of the process (Figure 8b,c). The global system resistance and energy gain could be due to the cumulative effect of scaling mitigation and lower ohmic resistance of the diffusion layer after the pause lapse. This allowed, for a short moment after current reestablishment, a concentration polarization higher than the one normally allowed by the actual limiting current density. This gain was more visible at higher frequencies such as the ones used in Sistat et al. in 2015 on model salt solution and Lemay et al. in 2019 on sweet whey [18,20].

### 3.3. Membrane Analysis

#### 3.3.1. Membrane Photographs

After each ED experiment, membrane photographs were taken to visually ascertain the presence of fouling, and more particularly scaling. For all PEF conditions tested, CEMs were free of fouling/scaling but regarding AEMs, as shown in Figure 9, the concentrate side of the membranes was affected by a certain amount of scaling depending on the PEF condition applied. A clear difference was visible between PEF conditions with increasing scaling intensity for higher pulse durations (15 s/25 s < 25 s/25 s < 35 s/25 s), shorter pause durations (25 s/35 s < 25 s/25 s < 25 s/15 s), and lower frequency (5 s/5 s < 15 s/15 s < 25 s/25 s < 35 s/35 s).

A previous study conducted in the laboratory [13] had shown the adverse impact of this AEMs scaling during acid whey ED on lactic acid removal and energy consumption and the beneficial influence of PEF application in its mitigation. Here, by judging the differences between the eight conditions tested, it seemed that conditions with longer pause and shorter pulse durations led to better scaling mitigation. Casademont et al. in 2009 [16] as well as Cifuentes-Araya et al. in 2011 [17] also demonstrated the beneficial impact of such PEF conditions on CEM scaling mitigation during model salt solutions ED. However, this beneficial impact during ED of complex solution such as acid whey has not yet been demonstrated. Moreover, it does not corroborate the conclusion reached by Cifuentes-Araya et al. in 2013 [33] where PEF conditions with shorter pause than pulse durations had better results regarding scaling mitigation during model salt solution ED. The experiments were also conducted at different scales and different current density (over-limiting current density) and solution flow rates that are key parameters influencing the establishment of membrane scaling which could explain the different results [34].

#### 3.3.2. Mineral Content

Analysis of the membranes’ mineral content allowed the determination of the composition and amount of scaling observed on the photographs. Regarding the CEMs that were not subjected to visible scaling, the mineral analysis did not show any significant difference between the eight PEF conditions tested for all mineral considered, see Figure 10. However, the concentrations of calcium and sodium before and after ED were statistically different (Figure 10a,c). Calcium concentration increased from an average 0.11 ± 0.00 g/100 g of membrane to 1.03 ± 0.16 g/100 g while sodium concentration dropped from 4.31 ± 0.05 g/100 g to 3.50 ± 0.16 g/100 g (P < 0.001). Taking into account the other minerals and experimental errors, the decreasing amount of sodium was replaced by calcium. 

Concerning the AEMs, the PEF conditions had a significant impact on the mineral concentrations, regardless of the mineral. Furthermore, the impact of PEF parameters was substantially similar for all the minerals observed. The results are reported in Figure 11 where conditions 25 s/35 s, 15 s/25 s 15 s/15 s, and 5 s/5 s were free of scaling with no statistical difference in terms of mineral concentration with the pristine membrane for all minerals (P > 0.05). In contrast, the application of conditions 25 s/15 s, 35 s/25 s, and 35 s/35 s resulted in a drastic increase of all minerals in the membranes (P < 0.001). Condition 25s/25s on the other hand, had the only significant amount of calcium and phosphorus in comparison with the pristine membrane. Focusing on pulse duration, pause duration, or frequency variations allowed us to highlight the increasing membrane mineral concentration with increasing pulse duration, decreasing pause duration, and lower frequency. The membranes’ mineral cumulative amount for each PEF condition is reported in Table 3 and compared with the pristine membrane. For conditions 35 s/25 s, 35 s/35 s, and 25 s/15 s, AEMs presented significant amounts of minerals (P < 0.01) while for the other PEF conditions no statistical difference with the pristine membrane was demonstrated. The same conclusion as for the specific mineral concentration can be drawn from the AEMs’ total mineral content with an increasing trend for longer pulse duration, shorter pause duration, and lower frequency conditions.

From above results, one can infer that there was no visible scaling or fouling on the CEMs after experiment, and the mineral analysis confirmed this. The change in calcium and sodium concentrations can be explained by the higher affinity for the membrane’s ion-exchange groups and longer retention duration of calcium compared to sodium as mentioned in Section 3.1.3 [28,29]. The minerals found here were either part of the membrane’s components or ions trapped in the membrane’s channels at the end of the ED treatment. Regarding AEMs, the scaling variations were the same as the ones from the membranes’ photographs. There were two types of conditions where scaling was greatly mitigated: 1) conditions with the highest frequencies (5 s/5 s and 15 s/15 s with 0.1 Hz and 0.03 Hz, respectively) allowing better benefit from the repeated diffusion layer reduction and solution concentration homogenisation during each pause duration [18,20] and 2) conditions with the lowest pulse/pause ratios (15 s/25 s and 25 s/35 s with ratios of 0.6 and 0.7, respectively) for which the improvements could be related to the longer pause duration allowing the solution flow to rinse the membrane’s surface and get rid of the freshly established calcium phosphate scaling [10,12,13]. Also, the amorphous form of calcium phosphate found on the membrane’s surface was a poorly soluble compound and its precipitation reaction can be partially reversed by the re-homogenisation of the OAComp solution [35]. Regarding the other PEF conditions, 25 s/25 s AEMs had only calcium and phosphorus concentrations higher than those of the pristine membrane while mineral concentrations for 25 s/15 s, 35 s/25 s, and 35 s/35 s conditions rose all together. As the scaling became more developed, its thickness increased and other minerals became trapped in it or participated in other scaling reactions [36]. The increasing amount of scaling could also explain the inability of the solution flow to wash out enough calcium phosphate during the pause lapses, the precipitates being already too strongly attached to the membranesurface.

## 4. Discussion

The application of different PEF conditions led to the improvement or decrease of the ED performances. Several criteria were particularly important to assess the prevalence of one condition over another. The global demineralisation and lactic acid removal from the acid whey were similar for all conditions (68.0 ± 0.5% and 44.5 ± 1.2%, respectively). It was therefore not possible to distinguish the advantages of the different PEF parameters studied, namely: pulse duration, pause duration, and pulse-pause frequency. However, the measurement of specific mineral removal, electrical resistance, and relative energy consumption as well as the scaling obtained for each of the eight PEF conditions allowed for the determination of significant impacts and tendencies regarding the above-mentioned PEF parameters. Moreover, taking into account the ED treatment duration for equivalent effective current application between the PEF conditions, which is an important factor to consider in an industrial application, profiling the results of the studied surface area was possible. As could be observed on Figure 12, for the same pause duration, an increasing pulse lapse led to higher resistance and energy consumption caused by an increasing scaling formation. By lengthening the pulse period, the concentration polarization gradient increased and reached the limiting current density faster, leading to water splitting and the establishment of pro-scaling conditions (alkaline pH in the membrane boundary on the OAComp side). On the other hand, the pulse duration variation did not consistently affect the removal of mineral ions, unlike the fluctuation of the pause duration. Indeed, increasing the duration of pause lapses led to higher divalent ions removal at the expense of sodium migration due to the higher affinity and longer retention time of calcium and magnesium ions by the CEM’s ion-exchange groups [28,29]. Long pauses also allowed for reduction of the AEM’s scaling due to a better exploitation of the solution flow rinsing effect of the membrane’s surface as well as homogenising ionic concentrations in the solutions, thus hampering the scaling formation and development. Finally, PEF frequency variation impacts have already been studied by several authors on model salt solutions [19,20] and more recently on sweet whey [18]. However, the beneficial impacts were only visible for high-frequency PEFs (higher than 1 Hz) in regard to mass transfer improvements. In this study, an overall enhancement of the ED process is visible through the increasing of frequency, even for frequencies between 0.014 and 0.1 Hz. For instance, the 5 s/5 s and 15 s/15 s conditions reached important scaling mitigation as well as low global system resistance and low relative energy consumption while displaying improved calcium and magnesium removal without critical reduction of sodium separation, especially for the 5 s/5 s condition. Incidentally, these conditions have a pulse/pause ratio of 1. The beneficial use of PEF conditions with such ratios has been observed by several authors for the treatment of model salt solution or even sweet whey with variable frequencies such as 0.05 Hz [17] and 5 Hz [18], respectively. In addition to the product treatment improvement, these conditions possess another advantage compared to the conditions with longer pause duration, which is the process length. Even though longer pauses have shown noticeable enhancements in scaling reduction, their process duration make their application less advantageous on an industrial scale. 

## 5. Conclusions

In this study, the global performance of ED in acid whey demineralisation and lactic acid removal, under eight PEF conditions, was evaluated. The chosen PEF conditions allowed for the determination of the influence of variations in the pulse/pause durations as well as of frequencies during ED treatment, and the comparison of these to the ones described in the literature. Scaling mitigation by use of a longer pause duration, the increased energy consumption for longer pulse duration, and the overall improvement of scaling reduction, mineral separation, and energy consumption for PEF conditions of higher frequencies are all described in this study and the associated mechanisms explained. Very few studies have been reported on complex food solutions in sub-limiting current density due to the complexity in composition and the different mechanisms present as well as their interactions to find the right explanation. Hence, the application of PEF in the treatment of such complex solutions is still in its early stages. Thus, the understanding of each parameter’s impact is extremely important in reducing trial and error waste of time during new industrial applications. In the same line of thought, the next step to support these results would be to conduct experiments at a larger scale to determine their consistency with different volumes, solution flows, or current density.

## Figures and Tables

**Figure 1 membranes-10-00014-f001:**
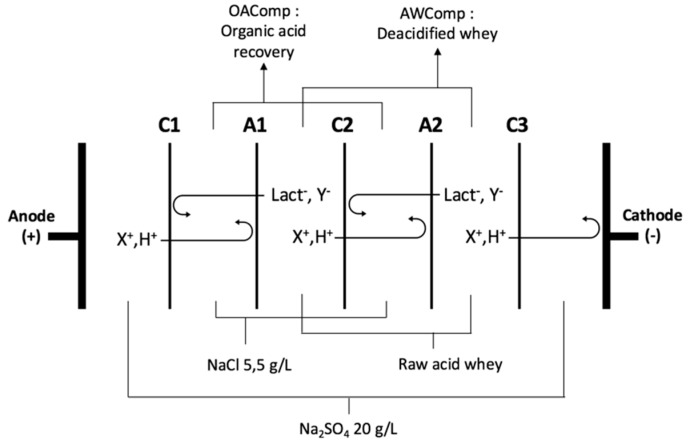
Electrodialysis (ED) configuration (CACAC, letters corresponding to the membrane’s stacking) used for acid whey deacidification. C refers to cation exchange membrane and A to anion exchange membrane. X+ and Y- refer to cations and anions, respectively, present in the whey. (modified from Dufton et al. 2018 [10]).

**Figure 2 membranes-10-00014-f002:**
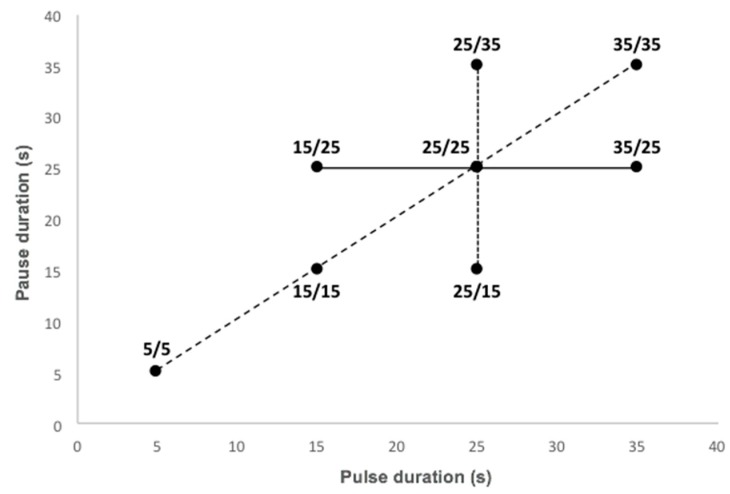
Surface area covered by the eight PEF conditions tested. The solid line, short dashes, and long dashes regroup conditions selected to study respectively pause duration, pulse duration, and frequency variation effect on acid whey electrodialysis.

**Figure 3 membranes-10-00014-f003:**
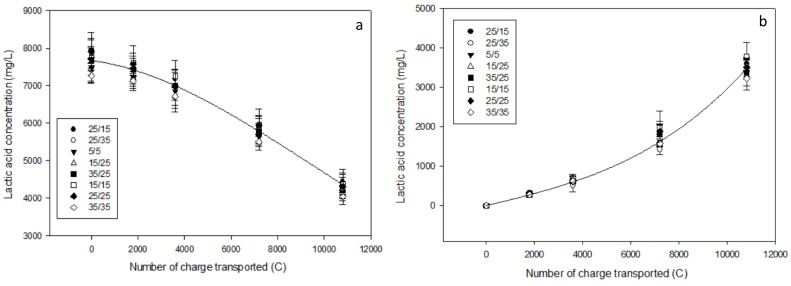
Evolution of lactic acid and lactate concentration (in mg/L) during electrodialysis in (**a**) the AWComp and (**b**) OAComp for all PEF conditions tested.

**Figure 4 membranes-10-00014-f004:**
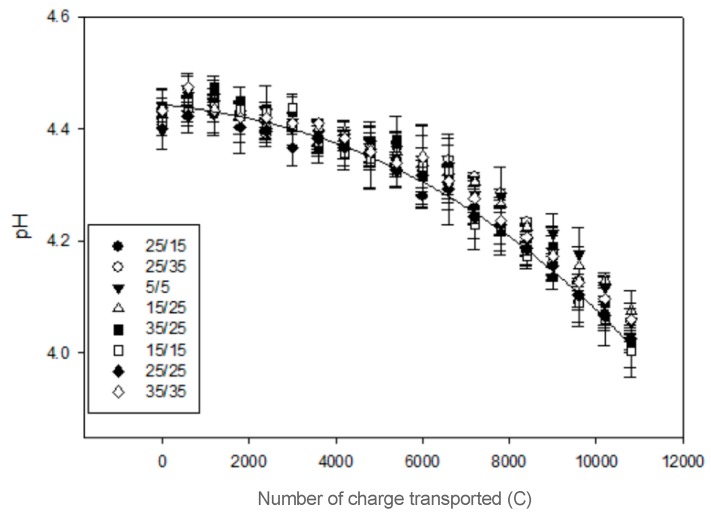
Acid whey pH evolution in the AWComp for the eight PEF conditions tested.

**Figure 5 membranes-10-00014-f005:**
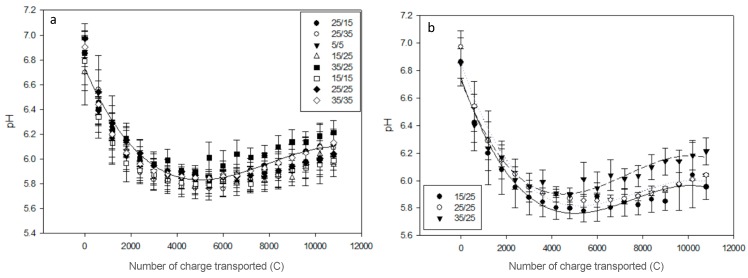
Concentrate solution pH evolution in (**a**) the OAComp for the eight PEF conditions tested and (**b**) for the three conditions representing the effect of pulsation time effect during PEF-ED treatment: 15 s/25 s, 25 s/25 s, and 35 s/25 s.

**Figure 6 membranes-10-00014-f006:**
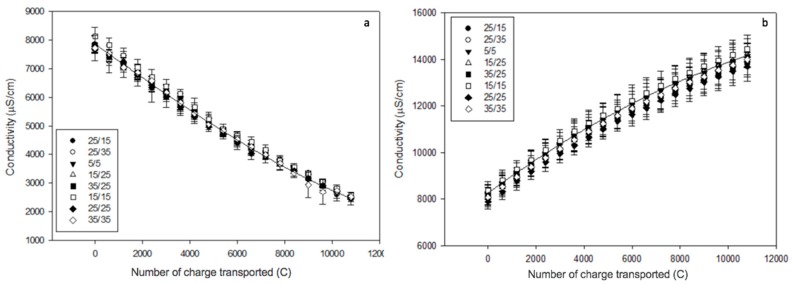
Conductivity evolution (**a**) in the AWComp and (**b**) OAComp for the eight PEF conditions tested.

**Figure 7 membranes-10-00014-f007:**
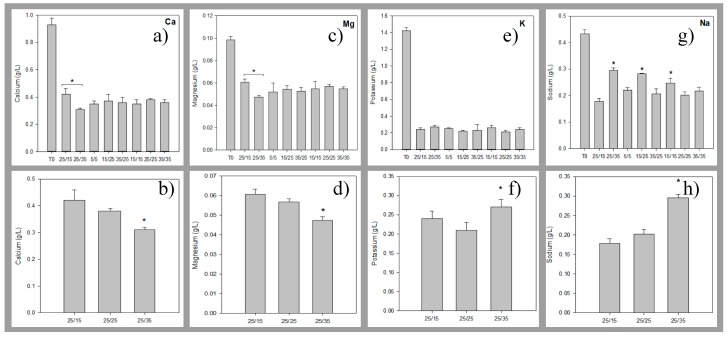
Mineral concentrations determined by inductively coupled plasma as atomisation and excitation source (ICP-OES) measurement (first line) for the eight PEF conditions tested and (second line) for the three conditions illustrating the pause duration variations during treatments: 25 s/15 s, 25 s/25 s, and 25 s/35 s. The minerals analysed were (**a**,**b**) calcium, (**c**,**d**) magnesium, (**e**,**f**) potassium, and (**g**,**h**) sodium. ** there is significant statistical difference with other conditions (P < 0.05).*

**Figure 8 membranes-10-00014-f008:**
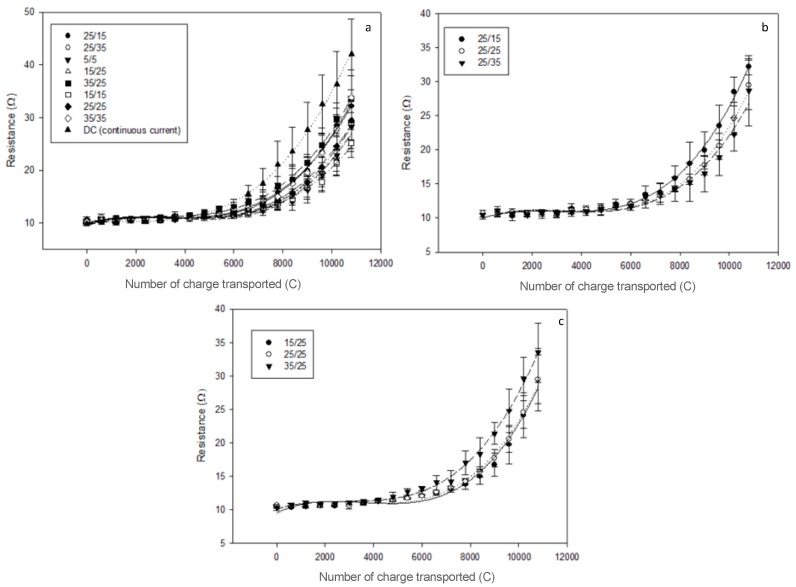
Global system resistance evolution for (**a**) the eight PEF conditions tested plus the application of continuous current from a previous study [13], (**b**) the three PEF conditions involving pause duration variations, and (**c**) the three involving pulse duration variations.

**Figure 9 membranes-10-00014-f009:**
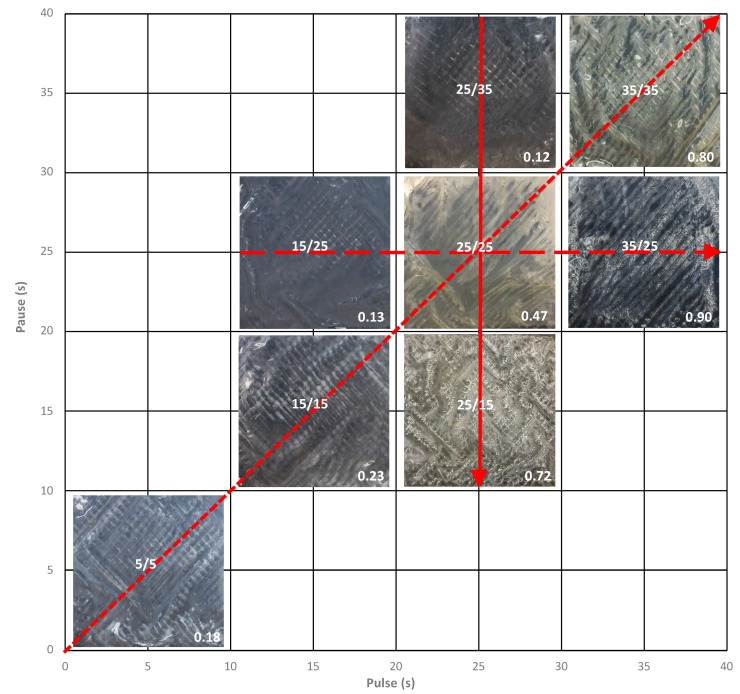
Anion-exchange membranes’ (AEMs’) concentrate side photographs after acid whey ED for the eight PEF treatments. The membranes represented are either A1 or A2 as there were no visible differences between them for the same condition. The scaling increasing intensity for PEF pause duration, pulse duration, and frequency variations are represented by the solid arrow, the long dots arrow, and the short dots arrow, respectively. The values in the corners are the amount of scaling on the AEMs (in g/100 g of dry membrane).

**Figure 10 membranes-10-00014-f010:**
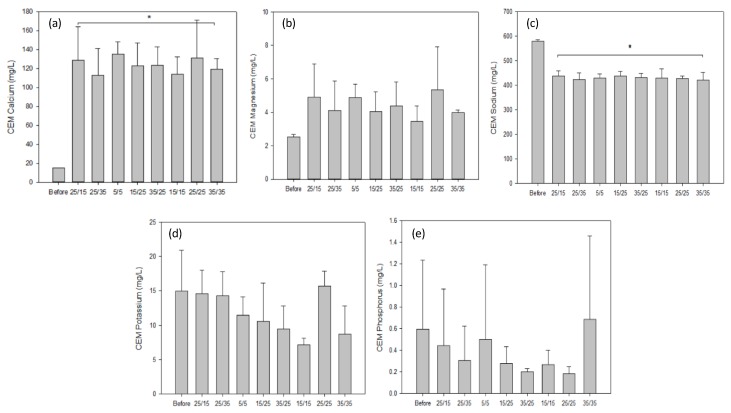
CEM mineral content for all PEF conditions tested. ICP-OES measurement was conducted after each ED run for (**a**) calcium, (**b**) magnesium, (**c**) sodium, (**d**) potassium, and (**e**) phosphorus and compared to the content of the pristine membrane. ** there is a statistical difference between these values and that of the pristine membrane (P < 0.05)*.

**Figure 11 membranes-10-00014-f011:**
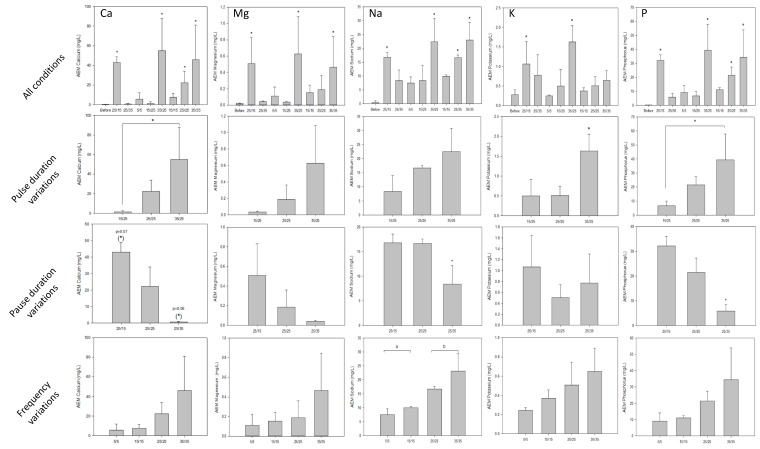
AEMs mineral content for all PEF conditions tested. ICP measurement was conducted after each ED run for calcium, magnesium, sodium, potassium, and phosphorus and compared to the content of pristine membranes or other conditions membrane. For all conditions, ** there is a statistical difference between these values and that of the pristine membrane (P < 0.05 with Dunnett test). For the parameter variations on rows, * or different letters on histograms means there is statistical difference between their value and the others (P < 0.05 with Tukey test).*

**Figure 12 membranes-10-00014-f012:**
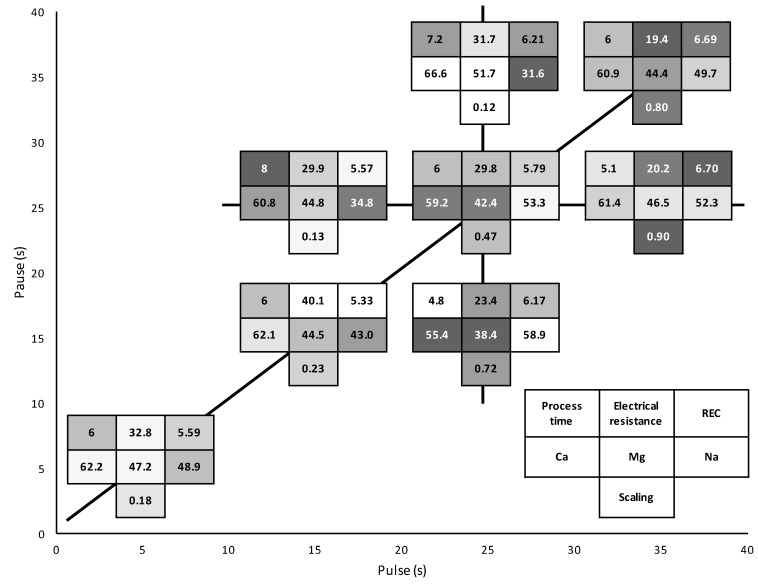
Schematic representation of the studied surface area comprised in the eight PEF conditions tested. Each condition is represented by a table including in the first row: its process duration (in h), its final electrical resistance reduction from the CC application results reported by [13] (in %), and its relative energy consumption (in Wh/g of lactic acid removed). The second row includes the demineralization rates of calcium, magnesium, and sodium (in %), while the third row reports the amount of scaling on the AEMs (in g/100 g of dry membrane). The shades of grey represent the position of the condition among the others: the lighter the grey, the better the result in term of acid whey treatment.

**Table 1 membranes-10-00014-t001:** Raw acid whey composition and physicochemical characteristics [2,5,9,10,13,21].

Characteristics	Unit	Acid Whey	Acid Whey from First Studies	Values Reported in the Literature
Total solids	g/L	57.5 ± 2.6	57.2 ± 1.5–59.8 ± 4.2	50.0–70.0
Lactose	g/L	36.7 ± 1.1	34.9 ± 1.0–41.2 ± 0.9	38.0–49.0
Minerals	g/L	6.1 ± 0.4	5.1 ± 1.1–6.9 ± 0.1	4.7–7.5
P	g/L	0.68 ± 0.03	0.55 ± 0.01–0.76 ± 0.02	0.44–0.90
Ca	g/L	0.93 ± 0.05	0.86 ± 0.02–1.08 ± 0.02	0.43–1.60
K	g/L	1.42 ± 0.04	1.26 ± 0.05–1.65 ± 0.03	1.28–1.82
Mg	g/L	0.10 ± 0.00	0.09 ± 0.00–0.10 ± 0.00	0.09–0.19
Na	g/L	0.43 ± 0.02	0.39 ± 0.03–0.53 ± 0.02	0.40–0.61
Lactate	g/L	7.67 ± 0.37	7.00 ± 0.14–7.12 ± 0.11	5.18–8.00
Ratio Lactate/Lactose	-	0.21	0.17–0.20	0.12–0.15
pH	-	4.4	4.4–4.6	4.0–4.6
Conductivity	mS/cm	7.81 ± 0.14	7.05 ± 0.24–7.09 ± 0.35	8.27 ± 0.42

**Table 2 membranes-10-00014-t002:** Relative energy consumption for the eight PEF conditions after 3 hours of effective current application during the ED treatment of acid whey.

Condition	Energy Consumption (Wh/g)
35 s/25 s	6.7 ± 1.0
35 s/35 s	6.7 ± 1.1
25 s/35 s	6.2 ± 1.3
25 s/15 s	6.2 ± 0.7
25 s/25 s	5.8 ± 0.3
5 s/5 s	5.6 ± 0.8
15 s/25 s	5.6 ± 0.3
15 s/15 s	5.3 ± 0.4

**Table 3 membranes-10-00014-t003:** AEM’s cumulative mineral content for a pristine membrane and the eight PEF conditions after 3 hours of effective current application during the ED treatment of acid whey.

Condition	Cumulative Mineral Content (g/100 g of Membrane)
35 s/25 s	0.90 ± 0.45 *
35 s/35 s	0.80 ± 0.47 *
25s/15 s	0.72 ± 0.08 *
25 s/25 s	0.47 ± 0.13
15 s/15 s	0.23 ± 0.04
5 s/5 s	0.18 ± 0.11
15 s/25 s	0.13 ± 0.08
25 s/35 s	0.12 ± 0.06
Before ED	0.01 ± 0.01

** there is a statistical difference between the PEF condition value and that of the pristine membrane (P < 0.05 with Dunnett test)*.

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
