# Peer review of "Systematic Study of the Impact of Pulsed Electric Field Parameters (Pulse/Pause Duration and Frequency) on ED Performances during Acid Whey Treatment"

_membranes, 2020, doi:10.3390/membranes10010014_

Round 1

Reviewer 1 Report

Comments to the Author The manuscript entitled “Systematic study of the impact of pulsed electric field parameters (pulse/pause duration and frequency) on ED performances during acid whey treatment” investigated the process optimization and the underlying mechanisms involved in PEF process improvements. The PEF parameters as pause duration, pulse duration and frequency were studied and to compare them to the ones described in the corresponding literatures. Through a thorough reading and reviewing, I believe this paper shows the good results, and the work is useful to the large scale experiment. However, the work will have more room to be improved if the details can be optimized as suggested below. 1. In the first paragraph of Introduction section, please give the reason why you choose the electrodialysis as the experimental method rather than ion-exchange resins and nanofiltration. 2. In the fourth paragraph of Introduction section, the authors mentioned n the recent years, these phenomena were successful controlled and mitigated by the application of pulsed electric fields (PEF) in a number of studies treating model salt solutions [15–17] and more recently in a few food solutions treatment [13,18], please add some simplified examples. 3. In the last paragraph of Introduction section, please describe your work more detailedly. 4. In the Electrodialytic configuration section, the performances of cation and anion exchange membranes should be described. 5. In the Protocol section, when did the experiment stopped. 6. In the pH section, the author claimed that regarding the OAComp, the pH decreased due to protons migrating from the acid whey and then stabilize before slowly increasing again because of water-splitting, whether the value of current efficiency could be provided here to illustrate the leakage of protons. 7. In the first paragraph of Global system resistance and relative energy consumption section, the reason why the global system resistance increases over time also needs to be explained. 8. In Table 2, why the energy consumption decreases when the PEF conditions are 15s/25s and 15s/15s.

Author Response

Responses to Reviewer 1

The manuscript entitled “Systematic study of the impact of pulsed electric field parameters (pulse/pause duration and frequency) on ED performances during acid whey treatment” investigated the process optimization and the underlying mechanisms involved in PEF process improvements. The PEF parameters as pause duration, pulse duration and frequency were studied and to compare them to the ones described in the corresponding literatures. Through a thorough reading and reviewing, I believe this paper shows the good results, and the work is useful to the large scale experiment.

However, the work will have more room to be improved if the details can be optimized as suggested below.

In the first paragraph of Introduction section, please give the reason why you choose the electrodialysis as the experimental method rather than ion-exchange resins and nanofiltration.

Done as requested

In the fourth paragraph of Introduction section, the authors mentioned n the recent years, these phenomena were successful controlled and mitigated by the application of pulsed electric fields (PEF) in a number of studies treating model salt solutions [15–17] and more recently in a few food solutions treatment [13,18], please add some simplified examples.

Done as requested.

In the last paragraph of Introduction section, please describe your work more detailedly.

Done as requested

In the Electrodialytic configuration section, the performances of cation and anion exchange membranes should be described.

This information is confidential

In the Protocol section, when did the experiment stopped.

As already mentioned in the Protocol section, «The experiment durations were calculated according to the pulse/pause combination applied for all conditions to correspond to an effective treatment of three hours or to a number of charge transported of 10800 C».

In the pH section, the author claimed that regarding the OAComp, the pH decreased due to protons migrating from the acid whey and then stabilize before slowly increasing again because of water-splitting, whether the value of current efficiency could be provided here to illustrate the leakage of protons.

Since, such a phenomenon was already observed and described in details by refs 10 and 13, and since no difference in term of pH trends amongst the different PEF conditions was observed, the current efficiency was not calculated.

In the first paragraph of Global system resistance and relative energy consumption section, the reason why the global system resistance increases over time also needs to be explained.

Explained as requested.

In Table 2, why the energy consumption decreases when the PEF conditions are 15s/25s and 15s/15s.

As already mentionned in the text, the differences are not statistically different, so it is not possible to conclude to a decrease. However, the energy gain would be due to the cumulative effect of scaling mitigation and lower ohmic resistance of the diffusion layer after the pause lapse

Reviewer 2 Report

In the manuscript, the acid whey was treated using electrodialysis with pulsed electric field. Eight PEF conditions of different pause and pulse durations were tested for process optimization and understanding the underlying mechanisms. The electric field of low frequencies were also investigated in sub-limiting current conditions. So I recommend the publication of this work. However, some issues within the manuscript need to be addressed before publication. 1. The meaning of the abbreviations such as “AOAC” in Page 4, Line 145 should be explained. 2. The reasons for pH decrease in Figure 4 need to be elaborated for better understanding. 3. As a suggestion, the leakage of H+ and OH- through AEM and CEM can be considered for the discussion on pH change in Page 8. 4. In Figure 7, the figures has to be numbered as a,b,c… corresponding to the caption.

Author Response

Responses to Reviewer 2 :

In the manuscript, the acid whey was treated using electrodialysis with pulsed electric field. Eight PEF conditions of different pause and pulse durations were tested for process optimization and understanding the underlying mechanisms. The electric field of low frequencies were also investigated in sub-limiting current conditions. So I recommend the publication of this work. However, some issues within the manuscript need to be addressed before publication.

The meaning of the abbreviations such as “AOAC” in Page 4, Line 145 should be explained.

Done as requested

The reasons for pH decrease in Figure 4 need to be elaborated for better understanding.

Explanations were added as requested.

As a suggestion, the leakage of H+ and OH- through AEM and CEM can be considered for the discussion on pH change in Page 8.

In very recent studies, it was demonstrated that this impact is less important when organic anions dissociation and water splitting phenomenon appeared.

In Figure 7, the figures has to be numbered as a,b,c… corresponding to the caption.

Done as requested.

Reviewer 3 Report

The authors conducted an experimental study of pulsed electric field parameters (pulse/pause duration and frequency) on ED performances during acid whey treatment. For the first time, pulsed electric field parameters of relatively low frequencies (< 1) were studied in sub-limiting current condition on a complex solution such as acid whey.

I would like to point out that perhaps the readers would also be interested in the impact of pulse and pause duration on voltage vs. time during application if PEF mode

The results are interesting and the manuscript is well written. The present form can be accepted for publication.

Author Response

Responses to Reviewer 3 :

The authors conducted an experimental study of pulsed electric field parameters (pulse/pause duration and frequency) on ED performances during acid whey treatment. For the first time, pulsed electric field parameters of relatively low frequencies (< 1) were studied in sub-limiting current condition on a complex solution such as acid whey.

I would like to point out that perhaps the readers would also be interested in the impact of pulse and pause duration on voltage vs. time during application if PEF mode.

We will keep in mind this interesting suggestion for our next works.

The results are interesting and the manuscript is well written. The present form can be accepted for publication.

Thanks a lot for your kind evaluation.

Round 2

Reviewer 1 Report

Revision is well done, and the work is of interest. Suggestion to accept.